



# Short-term Prediction of the Significant Wave Height and Average Wave Period based on VMD-TCN-LSTM Algorithm

Qiyan Ji[1], Lei Han[1] Lifang Jiang[2], Yuting Zhang[1], Minghong Xie[1] and Yu Liu[1,3]

[1]Marine Science and Technology College, Zhejiang Ocean University, Zhoushan 316022, China
[2]South China Sea Forecast and Disaster Reduction Center, Ministry of Natural Resources, Guangzhou 510000, China
[3]Southern Marine Science and Engineering Guangdong Laboratory (Zhuhai), Zhuhai 519000, China

*Correspondence to*: Lei Han (s20070700026@zjou.edu.cn)

**Abstract.** The present work proposes a prediction model of significant wave height (SWH) and average wave period (APD) based on variational mode decomposition (VMD), temporal convolutional networks (TCN), and long short-term memory
(LSTM) networks. The wave sequence features were obtained using VMD technology based on the wave data from the National Data Buoy Center. Then the SWH and APD prediction models were established using TCN, LSTM, and Bayesian hyperparameter optimization. The VMD-TCN-LSTM model was compared with the VMD-LSTM (without TCN cells) and LSTM (without VMD and TCN cells) models. The VMD-TCN-LSTM model has significant superiority and shows robustness and generality in different buoy prediction experiments. In the 3-hour wave forecasts, VMD primarily improved
the model performance, while the TCN had less influence. In the 12-, 24-, and 48-hour wave forecasts, both VMD and TCN improved the model performance. The contribution of the TCN to the improvement of the prediction result determination coefficient gradually increased as the forecasting length increased. In the 48-hour SWH forecasts, the VMD and TCN improved the determination coefficient by 132.5 % and 36.8 %, respectively. In the 48-hour APD forecasts, the VMD and TCN improved the determination coefficient by 119.7 % and 40.9 %, respectively.

## 1 Introduction

Ocean waves are crucial ocean physical parameters, and wave forecasts can significantly improve the safety of marine projects such as fisheries, power generation, and marine transportation (Jain et al., 2011; Jain and Deo, 2006). The earlier wave forecasting methods that emerged were semi-analytical and semi-empirical, including the Sverdroup-Munk-Bretscheider (SMB) (Bretschneider, 1957; Sverdrup and Munk, 1947) and Pierson-Neumann-James (PNJ) methods
(Neumann and Pierson, 1957). However, empirical methods cannot describe sea surface wave conditions in detail. The most widely used methods for wave forecasts are those of the third-generation wave models, including WAM (Wamdi, 1988), SWAN (Booij et al., 1999; Rogers et al., 2003), and WAVEWATCH III (Tolman, 2009). Nevertheless, numerical modelling methods must consume much computational resources and time (Wang et al., 2018).

Neural network methods achieve higher-quality forecasting results that are less time and computationally cost-consuming.
Several neural network methods have been widely used for wave forecasts, e.g., artificial neural networks (ANN) (Deo and



Naidu, 1998; Mafi and Amirinia, 2017; Kamranzad et al., 2011; Malekmohamadi et al., 2008; Makarynskyy, 2004), recurrent neural networks (RNN) (P. et al., 2020), and long short-term memory (LSTM) networks (Gao et al., 2021; Ni and Ma, 2020; Fan et al., 2020). The prediction model designed using neural network algorithms individually has poor generalization ability due to the strong non-stationarity and non-linear physical relationship of waves.

Signal decomposition methods are effective in extracting original data features. To further improve the prediction model performance, some researchers have developed hybrid models of signal decomposition and neural networks to forecast wave parameters. For example, empirical wavelet transform (EWT) (Karbasi et al., 2022), empirical mode decomposition (EMD) (Zhou et al., 2021; Hao et al., 2022), and singular spectrum analysis (SSA) (Rao et al., 2013). However, EMD and its extended algorithms suffer from mode confounding and sensitivity to noise (Bisoi et al., 2019), and wavelet
transforms methods lack adaptivity (Li et al., 2017). Variational mode decomposition (VMD) (Dragomiretskiy and Zosso, 2014) has overcome the disadvantages of EMD and is currently the most effective decomposition technique (Duan et al., 2022). Models combining VMD and neural networks are applied in forecasting various time series data. For example, stock price prediction (Bisoi et al., 2019), air quality index prediction (Wu and Lin, 2019), wind power prediction (Duan et al., 2022), runoff prediction (Zuo et al., 2020), and wave energy prediction (Neshat et al.,
2022; Jamei et al., 2022).

Recent studies have shown that temporal convolutional networks (TCN) outperform ordinary network models in handling time-series data in several domains, such as flood prediction (Xu et al., 2021), traffic flow prediction(Zhao et al., 2019), and dissolved oxygen prediction (Li et al., 2022a). The TCN cells can significantly capture the short-term local feature information of the sequence data, while the LSTM cells are adept at capturing the long-term dependence of the sequence
data. The wave data observed by the buoy contains both short-term features and long-term patterns of wave variability and is very well-suited for forecasting using a hybrid prediction model that includes the advantages of TCN and LSTM cells.

Hyperparameter optimization (HPO) for neural networks is commonly regarded as a black-box problem that avoids neural network problems such as overfitting, underfitting or incorrect learning rate values, which tend to occur in constructing deep learning models. The latest HPO techniques are grid search, stochastic search, and Bayesian optimization (BO), etc. BO
provides a better hyperparameter combination in a shorter time compared to traditional grid search methods (Rasmussen, 2004). It is more robust and less probable to be trapped in a local optima problem. Therefore, BO is the most widely used HPO algorithm, which has been applied to wave prediction models based on neural network algorithms (Zhou et al., 2022; Cornejo-Bueno et al., 2018).

Significant wave height (SWH) and average wave period (APD) are essential parameters in calculating wave power (De
Assis Tavares et al., 2020; Bento et al., 2021). Their forecasts need to consider the original characteristics of waves, short-term variability, and long-term dependence. Therefore, in the study, we used wave data from the National Data Buoy Center (NDBC) around the Hawaiian Islands to design a hybrid VMD-TCN-LSTM model to forecast SWH and APD, and the BO algorithm was used to obtain the most optimal hyperparameters for the network model.





The remaining sections of this paper are organized as follows. In Section 2, the data and pre-processing are described, and in
Section 3, the methodologies employed in the study are presented. In Section 4, the decomposition process of the wave series data, the overall structure of the prediction model and the hyperparameter optimization results are presented. Section 5 discusses the performance differences between the VMD-TCN-LSTM, VMD-LSTM, and LSTM models at various forecasting periods. Finally, Section 6 provides our conclusions.

## 2 Materials

### 2.1 Data source

Buoy measurements are the most common data source for wave parameter forecasts (Cuadra et al., 2016). The research used buoy data from the NDBC of the National Oceanic and Atmospheric Administration (NOAA) (https://www.ndbc.noaa.gov/). Each buoy provides measurements of SWH, mean wave direction (MWD), wind speed (WSPD), wind direction (WDIR), APD, dominant wave period (DPD), sea level pressure (PRES), gust speed (GST), air temperature (ATMP), and water
temperature (WTMP) at a resolution of 10 minutes to 1 hour. The dataset uses 99.00 to replace the missing values, but the resolution is still 1 hour for wave parameters data. Four NDBC buoys located in different directions around the Hawaiian Islands (Fig. 1) were used in the research, The statistics of the geographic location and the water depth parameters of the buoys are shown in Table 1.

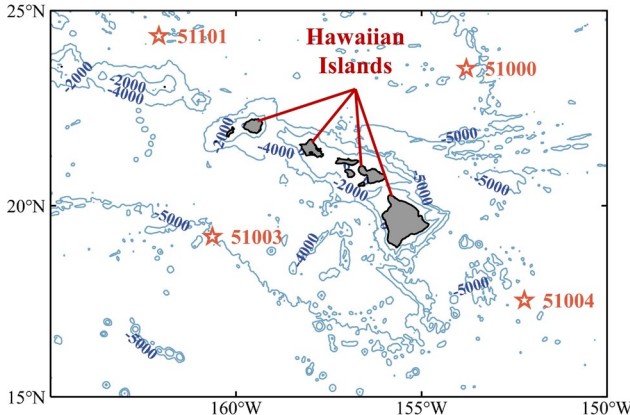

**Figure 1. The geographical locations of the 51000, 51003, 51004 and 51101 NDBC buoys.**

**Table 1. Statistics of the geographical locations and water depth parameters of the selected NDBC buoys.**

| Buoy ID | Latitude (°N) | Longitude (°W) | Depth (m) |
|---------|---------------|----------------|-----------|
| 51000 | 23.528 | 153.792 | 4762 |
| 51003 | 19.196 | 160.639 | 1987 |
| 51004 | 17.538 | 152.230 | 5278 |
| 51101 | 24.359 | 162.081 | 4860 |



## 2.2 Dataset partitioning and feature selection

Waves depend on previous wave height, sea surface temperature, sea temperature, wind direction, wind speed, and pressure
(Kamranzad et al., 2011; Nitsure et al., 2012; Fan et al., 2020). Because the buoy data have missing values, after data
filtering, the research selected data longer than two years at each buoy as the training datasets to capture the year-round
characteristics of wave parameters. The divisions and statistical characteristics of the training and testing datasets for the four
buoys are shown in Table 2 and Fig. 2.

**Table 2. NDBC datasets division and statistical information.**

| Buoy ID | Dataset | Date (YYYY/MM/DD HH) | Data volume | SWH range (m) | ADP range (s) |
|---------|---------|----------------------|-------------|---------------|---------------|
| 51000 | Training | 2015/08/20 22~2020/04/20 15 | 40594 | [0.89, 11.03] | [4.60, 14.89] |
| | Testing | 2020/07/29 01~2020/10/19 08 | 1976 | [0.89, 3.49] | [5.12, 11.43] |
| 51003 | Training | 2015/08/25 03~2018/08/07/13 | 25494 | [0.85, 6.83] | [4.75, 13.85] |
| | Testing | 2018/11/10 00~2018/12/31 08 | 1214 | [1.41, 4.70] | [5.48, 12.79] |
| 51004 | Training | 2014/07/06 14~2017/10/08 16 | 28400 | [0.86, 5.80] | [4.78, 14.03] |
| | Testing | 2017/12/08 11~2018/02/14 09 | 1631 | [1.29, 5.30] | [5.46, 12.99] |
| 51101 | Training | 2014/11/04 22~2018/03/29 07 | 29548 | [0.83, 8.54] | [4.46, 14.73] |
| | Testing | 2019/10/25 07~2020/01/04 00 | 1698 | [1.16, 6.04] | [5.44, 13.03] |

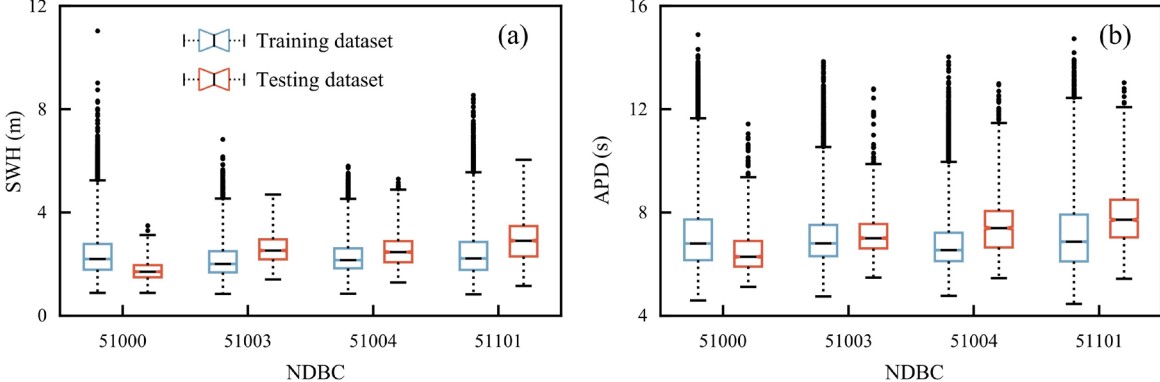


**Figure 2. Statistical analysis of SWH and APD on the training and testing datasets of the four NDBC buoys.**

The research selected SWH and APD, two wave parameters, as forecasting variables. The correlation between various
environmental parameters with SWH and APD was determined by calculating Pearson correlation coefficients between the
above parameters before selecting the input features. For the parameters $X$ and $Y$, the Pearson correlation coefficients are
calculated as follows.

$$r = \frac{cov(X, Y)}{\sigma_X \sigma_Y} = \frac{\frac{1}{n}\sum_{i=1}^{n}(X_i - \bar{X})(Y_i - \bar{Y})}{\sqrt{\frac{1}{n}\sum_{i=1}^{n}(X_i - \bar{X})^2}\sqrt{\frac{1}{n}\sum_{i=1}^{n}(Y_i - \bar{Y})^2}}, \tag{1}$$



The Pearson correlation coefficients between SWH, MWD, WSPD, GST, WDIR, PRES, WTMP, ATMP, APD, and DPD were calculated after neglecting the parameter values at unrecorded moments (Fig. 3). As shown in Fig. 3, SWH has a positive correlation with APD, DPD, MWD, WSPD, GST, WDIR, and PRES to different degrees, and SWH has a negative

correlation with WTMP and ATMP. Among them, WSPD and GST have a strong correlation ($r$=0.988), WTMP and ATMP have a strong correlation ($r$=0.901), and APD is considered to contain the main features of DPD. In order to utilize as many features of different physical parameters as possible while minimizing the computational redundancy, seven physical parameters, SWH, APD, MWD, WSPD, WDIR, PRES and ATMP, were selected as input and training data for SWH and APD forecasting in the study.

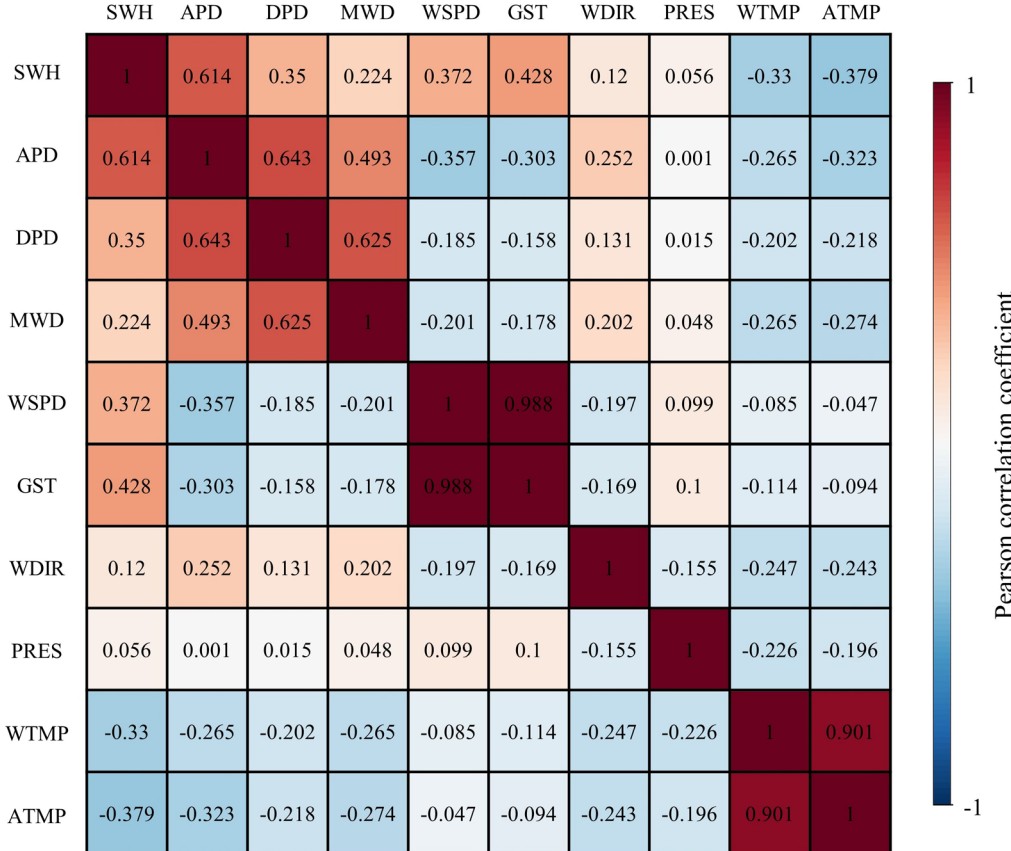


**Figure 3. Pearson correlation coefficients between various physical parameters in NDBC data.**

**2.3 Data pre-processing**

Wind and wave directions are continuous in space but discontinuous numerically. For example, the directions 2° and 358° are very close, but the magnitude of the values differs significantly. Therefore, the wind and wave directions need to be pre-

processed. The following formula recalculates the wind and wave directions (Nitsure et al., 2012).



$$\psi = \begin{cases} 1 - \frac{\theta}{180}, & if\ 0° \le \theta \le 180° \\ \frac{\theta-180}{180}, & if\ 180° < \theta < 360° \end{cases},$$ (2)

where $\theta$ is the original wind or wave directions and $\psi$ is the re-encoded value of wind or wave directions. $\psi$ has a range of values from [0, 1].

Since different NDBC physical variables have different units and magnitudes, this can substantially influence the
performance of the neural network model. Therefore, each variable must be normalized or standardized before using it as input data for the model (Li et al., 2022b). The research used a min-max normalization function to scale the input data between [0, 1], which is calculated as follows.

$$x_n = \frac{x - \min(x)}{\max(x) - \min(x)},$$ (3)

where $x_n$ is the normalized feature value and $x$ is the measured feature value.

**3 Methods**

**3.1 Variational mode decomposition (VMD)**

The VMD is an adaptive, completely nonrecurrent mode variation and signal processing technique that combines the Wiener filter, the Hilbert transform, and the Alternating Direction Method of Multipliers (ADMM) technique (Dragomiretskiy and Zosso, 2014). VMD can determine the number of mode decompositions for a given sequence according to the situation. It
has resolved the issues of mode mixing, boundary effects of EMD. The VMD decomposes the original sequence signal into an Intrinsic Mode Function (IMF) of finite bandwidth, where the frequencies of each mode component $u_k$ are concentrated around a central frequency $\omega_k$.

The nucleus of VMD is the construction and solution of the variational problem, which is essentially a constrained optimization problem. The variational problem is to minimize the sum of the estimated bandwidths of the IMFs, with the
constraint that the sum of the IMFs is the original signal. The calculation formula is as follows.

$$\min_{\{u_k\},\{\omega_k\}} \left\{ \sum_k \left\| \partial_t \left[ \left( \delta(t) + \frac{j}{\pi t} \right) * u_k(t) \right] e^{-j\omega_k t} \right\|_2^2 \right\}\quad s.t. \sum_k u_k = f,$$ (4)

where "$s.t.$" is the abbreviation of "subject to". $\{u_k\} := \{u_1, u_2, \dots, u_k\}$ and $\{\omega_k\} := \{\omega_1, \omega_2, \dots, \omega_k\}$ denote the set of all modes and their corresponding central frequencies, respectively. The $f$ is the original signal, $k$ is the total number of modes, and $\delta(t)$ represents the Dirac distribution. The $j$ is an imaginary unit and "*" denotes the convolution.
To simplify the above equations, VMD introduces a quadratic penalty term ($\alpha$) and Lagrange multipliers ($\lambda$) to convert the constrained problem into a non-constrained problem. $\alpha$ guarantees the reconstruction accuracy of the signal, and $\lambda$ maintains the constraint stringency.



$$\mathcal{L}(\{u_k, \omega_k\}, \lambda) := \alpha \sum_k \left\| \partial_t [(\delta(t) + \frac{j}{\pi t}) * u_k(t)] e^{-j\omega_k t} \right\|_2^2 + \left\| f(t) - \sum_k u_k(t) \right\|_2^2 + \langle \lambda(t), f(t) - \sum_k u_k(t) \rangle , \tag{5}$$

Finally, the ADMM solves the saddle point of the augmented Lagrange multiplier. Update the iterative formulas for $u_k$, $\omega_k$
and $\lambda$ as follows.

$$\hat{u}_k^{n+1}(\omega) = \frac{\hat{f}(\omega) - \sum_{i \neq k} \hat{u}_i(\omega) + \frac{\hat{\lambda}(\omega)}{2}}{1 + 2\alpha(\omega - \omega_k)^2} , \tag{6}$$

$$\omega_k^{n+1} = \frac{\int_0^\infty \omega |\hat{u}_k(\omega)|^2 d\omega}{\int_0^\infty |\hat{u}_k(\omega)|^2 d\omega} , \tag{7}$$

$$\hat{\lambda}^{n+1}(\omega) = \hat{\lambda}^n(\omega) + \tau \left( \hat{f}(\omega) - \sum_{k=1}^K \hat{u}_k^{n+1}(\omega) \right) , \tag{8}$$

where $\hat{f}(\omega)$, $\hat{u}_k(\omega)$, $\hat{\lambda}(\omega)$ and $\hat{u}_k^{n+1}(\omega)$ are the Fourier transforms of $f(\omega)$, $u_k(\omega)$, $\lambda(\omega)$ and $u_k^{n+1}(\omega)$, respectively. The $n$ and $\tau$ are
the number of iterations and update coefficients of Dual ascent. The iterations are stopped when the convergence condition
satisfies the following equation.

$$\sum_k \frac{\left\| \hat{u}_k^{n+1} - \hat{u}_k^n \right\|_2^2}{\left\| \hat{u}_k^n \right\|_2^2} < \epsilon , \tag{9}$$

The VMD algorithm can be found in more detail in Dragomiretskiy and Zosso (2014).

## 3.2 Temporal convolutional networks (TCN)

The TCN is a variant of the Convolutional Neural Networks (CNN) (Fig. 4). TCN model uses causal convolution, dilated
convolution, and residual block to extract sequence data with a large receptive field and temporality (Yan et al., 2020). TCN
performs convolution in the time domain (Kok et al., 2020), which has a more lightweight network structure than CNN,
LSTM, and GRU (Bai et al., 2018). TCN has the following advantages: (1) causal convolution prevents the disclosure of
future information, (2) dilated convolution extends the receptive field of the structure, and (3) residual block maintains the
historical information for a longer period. TCN is on the concept of causal convolution, where "causal" indicates that the
output $y_t$ at the time $t$ is only dependent on the input $x_1$, $x_2,\ldots,x_t$ and is not influenced by $x_{t+1}, x_{t+2},\ldots,x_T$. The receptive field
depends on the filter size and the network depth. However, the increase of filter size and network depth brings the risk of
gradient disappearance and explosion. To avoid these problems, TCN introduces dilated convolution based on causal
convolution (Zhang et al., 2019).



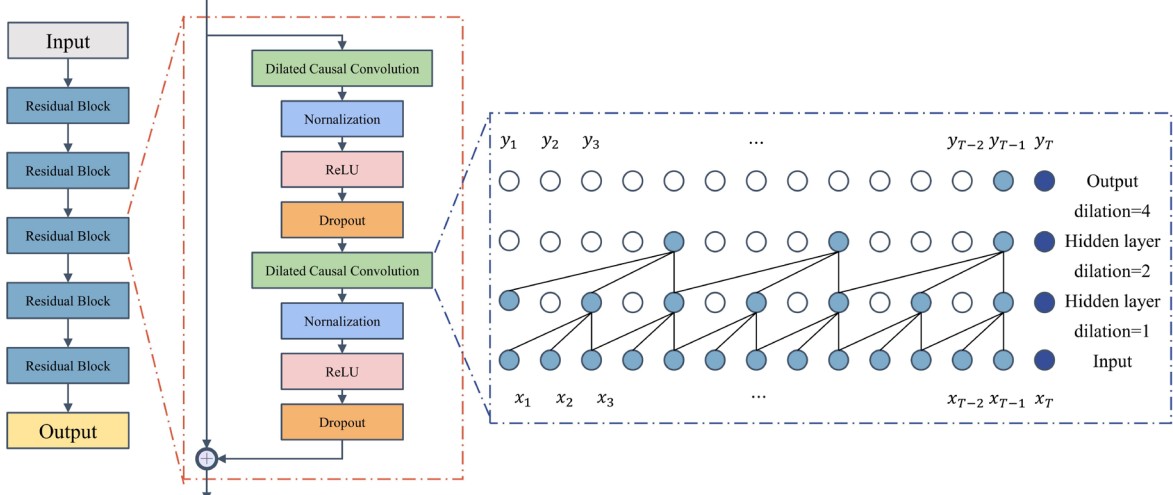


**Figure 4. Structure of temporal convolutional networks.**

TCN employs dilated convolution based on causal convolution to handle the linear increase in network depth for longer inputs. The dilated convolution introduces a dilation factor ($d$) to adjust the receptive field. The processing capability of long sequences depends on the filter size ($k$), $d$, and network depth ($p$). For a one-dimensional time series $x \in \mathbb{R}^n$ and a filter

$f$: $\{0, \ldots, k-1\} \rightarrow \mathbb{R}$, the dilated convolution operation $F$ on the sequence element $s$ is defined as:

$$F(s) = (x *_d f)(s) = \sum_{i=0}^{k-1} f(i) \cdot x_{s-d \cdot i} , \tag{10}$$

where $d$ is the dilation factor, $k$ is the filter size, and "$s - d \cdot i$" describes the passed direction. In the dilated convolution, the dilation factor $d$ grows exponentially ($d = 2^i$) with the hidden layer depth ($i$), and the receptive field is $(k - 1)d$. TCN effectively increases the receptive field without additional computational cost by increasing the dilation factor. Figure 4

illustrates the structural components of TCN with the dilation factors $d = 1$, 2, and 4.

To ensure training efficiency, TCN introduces multiple residual blocks to accelerate the prediction model. Each residual block comprises two dilated causal convolution layers with the same dilation factor, normalization layer, ReLU activation and dropout layer. The input of each residual block is also added to the output when the input and output channels are different (Fig. 4).

The input of each residual block is also added to the output when the number of channels between the input and output are different (Fig. 4). The following equation obtains the residual block:

$$o = Activation(x + F(x)) , \tag{11}$$

where $o$ is the residual block output, *Activation* is the activation function, $x$ is the previous input information, and $F(x)$ is the transformed information.





## 3.3 Long short-term memory (LSTM) networks

The traditional RNN is exposed to gradient explosion and vanishing risk. LSTM network learns to reset itself at the appropriate time by adding a forgetting gate in RNN, which releases internal resources. Meanwhile, LSTM learns faster by adding the self-looping method to generate a long-term continuous flow path. As a specific RNN, the LSTM network structure includes an input layer, a hidden layer, and an output layer. The structure of the LSTM cell is shown in Fig. 5.

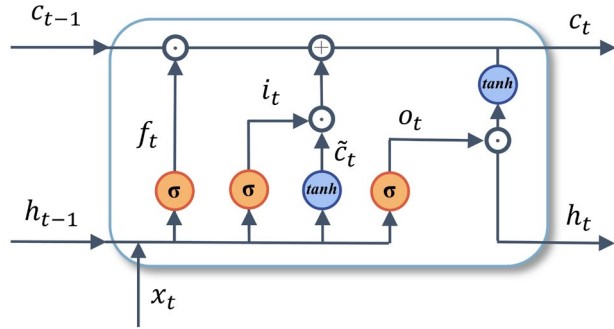

**Figure 5. Structure of long short-term memory networks.**

A LSTM cell consists of four components, the forget gate $f_t$, the input gate $i_t$, the storage cell state $c_t$ and the output gate $o_t$.

The $f_t$ determines the number of memories that need to be reserved from $c_{t-1}$ to $c_t$.

$$f_t = \sigma(W_f \cdot [x_t, h_{t-1}] + b_f) \,, \tag{12}$$

The $i_t$ determines the information that is input to this cell state.

$$i_t = \sigma(W_i \cdot [x_t, h_{t-1}] + b_i) \,, \tag{13}$$

The $o_t$ represents the information output from this cell state.

$$o_t = \sigma(W_o \cdot [x_t, h_{t-1}] + b_o) \,, \tag{14}$$

The cell state is:

$$C_t = f_t \odot C_{t-1} + i_t \odot \widetilde{C}_t \,, \tag{15}$$

$$\widetilde{C}_t = \tanh(W_c \cdot [x_t, h_{t-1}] + b_c) \,, \tag{16}$$

The next cell with $h_t$ is:

$$h_t = o_t \odot \tanh(C_t) \,, \tag{17}$$

In the above equation, $x_t$ denotes the current input vector, and $W$ and $b$ denote the hyperparameters of the weights and biases. The $h_t$ is the storage cell value at time $t$. The $\sigma$ is the sigmoid function, $tanh$ denotes the hyperbolic tangent function, "·" denotes the dot product of matrices, and "$\odot$" denotes the Hadamard matrix product of equidimensional matrices (Yu et al., 2019; Gers et al., 2000; Hochreiter and Schmidhuber, 1997). The sigmoid function takes values in the range is [0, 1], and in the forgetting gate, if the value is 0, the information of the previous state is completely forgotten, and if the value is 1, the information is completely retained. tanh function takes the values in the range [-1, 1].



## 3.4 Bayesian optimization (BO)

The BO aims to find the global maximizer (or minimizer) of the unknown objective function $f(x)$ (Frazier, 2018), as shown in follows.

$$x^* = (\arg \max_{x \in D} f(x)),\tag{18}$$

where $D$ denotes the search space of $x$, where each dimension is a hyperparameter.

The BO has two critical components, first, build an agency model of the objective function through a regression model (e.g., Gaussian process regression) and subsequently use the acquisition function to decide where to sample next (Frazier, 2018).

The Gaussian process (GP) is an extension of multivariate Gaussian distribution into an infinite dimensional stochastic process (Frazier, 2018; Brochu et al., 2010), which is the prior distribution of stochastic processes and functions. Any finite subset of random variables has a multivariate Gaussian distribution, and a GP is entirely defined by its mean function and covariance function (Rasmussen, 2004). BO optimizes the unknown function $f(x)$ by combining the prior distribution of the function based on the GP with the current sample information to obtain the posterior of the function.

The BO uses the expected improvement (EI) function as the acquisition function to evaluate the utility of the model posterior to determine the next input point. Let $f_n^*$ be the optimal value of the acquisition function at the current iteration. The EI acquisition function can be defined by Eq. (19) (Shahriari et al., 2016; Frazier, 2018):

$$EI(x) = \begin{cases} (\mu(x) - f_n^*)\phi(Z) + \sigma(x)\varphi(Z) & if\ \sigma(x) > 0 \\ 0 & if\ \sigma(x) = 0 \end{cases},\tag{19}$$

where $Z = \frac{\mu - f(x^+)}{\sigma(x)}$, $\mu(x)$ and $\sigma(x)$ are the expectation and variance of the input value $x$, respectively, $\phi$ is the cumulative distribution function (CDF) of the standard normal distribution, and $\varphi$ is the probability density function of the standard normal distribution.

The BO employs GP and EI in the iterations to evaluate and obtain the global optimal hyperparameters (Zhang et al., 2020a).

The framework of the Bayesian parameter optimization algorithm is shown below.

| **Algorithm 1** Basic pseudo-code for Bayesian optimization |
|---|
| 1: Initializing the prior distribution of the substitution function based on GP. |
| 2: **for** $n = 1, 2, \dots$ **do** |
|     Find $x_n$ by optimizing acquisition function. |
|     $x_{n+1} = \arg \max_x \alpha_{EI}(x\|D_n)$ |
|     Query objective function to obtain $y_{n+1}$ |
|     Augment data $D_{n+1} = \{D_n,\ (x_{n+1}, y_{n+1})\}$ |
|     Update the prior distribution of the substitution function. |
| **end for** |
| 3: Find the global optimal solution for the current GP. |



# 4 Wave parameter prediction model framework and parameter settings

## 4.1 Data decomposition and parameter setting

The input to the VMD method requires the original signal $f(t)$ and a predefined parameter $K$. The $K$ determines the number of IMF patterns extracted during the decomposition. If the number of the extracted patterns is too large, it leads to a decrease in
accuracy and unnecessary computational overhead (Liu et al., 2020). However, if the number of patterns is too small, the information in the patterns is insufficient to construct a high-precision prediction model. Therefore, it is essential to choose an appropriate one for $K$.

There is still a lack of general guidelines for the selection of the $K$ parameter (Bisoi et al., 2019). Methods commonly used in other fields include the central frequency observation method (Hua et al., 2022; Chen et al., 2022; Fu et al., 2021), sample
entropy (Zhang et al., 2020b; Niu et al., 2021), genetic algorithm (Huang et al., 2022), effective kurtosis index (Li et al., 2020), signal energy (Liu et al., 2020; Huang and Deng, 2021), etc. The central frequency observation method is convenient and effective, and it is used in this research to determine the number of patterns $K$ for sequence decomposition. For various $K$ parameter values, when the central frequency of the last mode has no significant changing trend, the number of $K$ currently is the optimal number of mode decompositions. The search range of $K$ parameters in the research is [5, 15], and Table 3
calculates the central frequency of the last mode after the SWH and APD were decomposed with different $K$ parameters. As shown in Table 3, after VMD decomposes SWH, the central frequency of the last mode from $K = 13$ does not change significantly, so the optimal VMD decomposition mode number for SWH is 13. As shown in Table 3, after VMD decomposes APD, the central frequency of the last mode from $K = 12$ does not change significantly, so the optimal VMD decomposition mode number for APD is 12. According to the optimal $K$ parameters of SWH and APD decomposition, the
SWH and APD data on the training and test sets of each buoy are decomposed by VMD separately.

Table 3. The central frequency of the last mode after SWH and APD decomposition with different K parameters.

| $K$ | 5 | 6 | 7 | 8 | 9 | 10 | 11 | 12 | 13 | 14 | 15 |
|---|---|---|---|---|---|---|---|---|---|---|---|
| last mode after the SWH were decomposed ($\times$ (1e – 6) Hz) | 395975 | 419471 | 447343 | 449966 | 452286 | 452888 | 453510 | 453650 | 476397 | 477347 | 477403 |
| last mode after the APD were decomposed ($\times$ (1e – 6) Hz) | 394750 | 433368 | 433725 | 434455 | 451044 | 451505 | 451713 | 475568 | 476213 | 476246 | 476317 |



## 4.2 Wave parameter prediction model framework

The overall structure of the VMD-TCN-LSTM wave parameter prediction model in the research is shown in Fig. 6,
including three parts: data pre-processing, VMD data decomposition, and model training and forecasting. TCN cells and
LSTM cells are used in the model to construct an encoder-decoder network with an attention mechanism. To evaluate the
accuracy of the VMD-TCN-LSTM model. The effect of the VMD technique and TCN cells on the forecasting results was
also analysed. The results of the VMD-TCN-LSTM model were compared with the VMD-LSTM and LSTM models. The
VMD-LSTM model used both LSTM cells for encoding and decoding. The LSTM model without the VMD technique for
data decomposition and was not encoded using TCN cells.

**Figure 6. The overall structure of VMD-TCN-LSTM wave parameter prediction model.**



### 4.3 Neural network hyperparameter optimization based on BO

In the research, the BO algorithm is used to search for the optimal hyperparameters for the training of the VMD-TCN-LSTM model, including batch size, number of TCN hidden layer units, number of LSTM hidden layer units, number of Dense hidden layer units, learning rate ($\alpha$), dropout rate, and L2 regularization parameter of LSTM layer. The hyperparameters search range and optimal results are shown in Table 4. Meanwhile, the learning rate decay and Early Stopping method are used to prevent overfitting of the model and reduce the wasted training time.

**Table 4. Bayesian hyperparameter optimization results.**

| Parameter | Search interval | Final value |
|---|---|---|
| Batch size | [8, 512] | 256 |
| TCN hidden unit | [16, 256] | 64 |
| LSTM hidden unit | [16, 256] | 128 |
| Dense hidden unit | [16, 256] | 128 |
| learning rate α | [1e-4, 1e-2] | 3e-4 |
| Dropout rate | [0.1, 0.5] | 0.2 |
| L2 regularization parameter | [1e-7, 1e-4] | 1e-5 |

## 5 Experiment and analysis

### 5.1 Evaluation metrics

To quantify the prediction model performance, the mean absolute error (MAE), root mean square error (RMSE), mean absolute percentage error (MAPE) and the determination coefficient ($R^2$) are used as evaluation metrics. The equations can be written as follows.

$$MAE = \frac{1}{N}\sum_{i=1}^{N}\left|y_{p(i)} - y_{t(i)}\right|, \tag{20}$$

$$RMSE = \sqrt{\frac{1}{N}\sum_{i=1}^{N}\left(y_{p(i)} - y_{t(i)}\right)^2}, \tag{21}$$

$$MAPE = \frac{1}{N}\sum_{i=1}^{N}\frac{\left|y_{p(i)} - y_{t(i)}\right|}{y_{t(i)}} \times 100\%, \tag{22}$$

$$R^2 = 1 - \frac{\sum_{i=1}^{N}\left(y_{p(i)} - y_{t(i)}\right)^2}{\sum_{i=1}^{N}\left(y_{t(i)} - \bar{y}_t\right)^2}, \tag{23}$$

where $N$ denotes the time length of the series data, $y_{t(i)}$ is the true observation values of NDBC, $y_{p(i)}$ is the predicted value,

and $\bar{y}_t$ is the average of the true observation values.

Furthermore, to quantify the improvement of the VMD technique and the TCN unit on the model accuracy, respectively, four parameters, $I_{MAE}$, $I_{RMSE}$, $I_{MAPE}$ and $I_{R^2}$ (Eqs. (24) to (27)), are introduced to compare the percentage improvement of the evaluation metrics of VMD-LSTM and VMD-TCN-LSTM models concerning the LSTM model.



$$I_{MAE} = \frac{MAE_{LSTM} - MAE_{model}}{MAE_{LSTM}} \times 100\% , \tag{24}$$

$$I_{RMSE} = \frac{RMSE_{LSTM} - RMSE_{model}}{RMSE_{LSTM}} \times 100\% , \tag{25}$$

$$I_{MAPE} = \frac{MAPE_{LSTM} - MAPE_{model}}{MAPE_{LSTM}} \times 100\% , \tag{26}$$

$$I_{R^2} = \frac{R^2_{model} - R^2_{LSTM}}{R^2_{LSTM}} \times 100\% , \tag{27}$$

where the subscript "*LSTM*" represents the evaluation metrics of the LSTM model, and the subscript "*model*" represents the evaluation metrics of the VMD-LSTM or VMD-TCN-LSTM models.

**5.2 3-hour forecasting performance**

The evaluation metrics of SWH and APD for different prediction models on the testing sets of the four buoys for the 3-hour forecasts are shown in Table 5, where the best results are shown in bold. As shown in the table, both the VMD-LSTM and VMD-TCN-LSTM models significantly outperform the results of the LSTM model. This indicates that the data pre-processing method of VMD can extract the features of the sequence data well in the 3-hour SWH and APD forecasts, which

can significantly improve the forecasting performance. Meanwhile, the improvement of the TCN cells on the model performance is not particularly significant in the 3-hour SWH and APD forecasts. The performance of the VMD-TCN-LSTM model was slightly better than that of the VMD-LSTM model only in some instances.

**Table 5. Accuracy evaluation of the three models in 3-hour SWH and APD forecasts.**

| Buoy ID | Algorithm | SWH | | | | APD | | | |
|---------|-----------|-----|-----|------|-----|-----|-----|------|-----|
| | | MAE (m) | RMSE (m) | MAPE (%) | $R^2$ | MAE (s) | RMSE (s) | MAPE (%) | $R^2$ |
| 51000 | VMD+TCN+LSTM | **0.083** | **0.111** | **4.675** | **0.924** | **0.155** | **0.204** | **2.353** | **0.950** |
| | VMD+LSTM | 0.085 | 0.118 | 4.766 | 0.918 | 0.168 | 0.217 | 2.536 | 0.942 |
| | LSTM | 0.143 | 0.177 | 8.527 | 0.874 | 0.219 | 0.290 | 3.295 | 0.897 |
| 51003 | VMD+TCN+LSTM | **0.066** | **0.082** | **2.315** | **0.978** | **0.105** | **0.145** | **1.438** | 0.973 |
| | VMD+LSTM | 0.067 | 0.088 | 2.592 | 0.976 | 0.108 | 0.147 | 1.479 | **0.976** |
| | LSTM | 0.153 | 0.204 | 5.862 | 0.869 | 0.252 | 0.353 | 3.397 | 0.859 |
| 51004 | VMD+TCN+LSTM | **0.080** | **0.105** | **2.816** | **0.973** | **0.115** | **0.158** | **1.323** | **0.981** |
| | VMD+LSTM | 0.081 | 0.107 | 2.856 | 0.970 | 0.124 | 0.164 | 1.372 | 0.976 |
| | LSTM | 0.159 | 0.217 | 6.105 | 0.885 | 0.279 | 0.393 | 3.601 | 0.884 |
| 51101 | VMD+TCN+LSTM | **0.093** | **0.124** | **4.720** | 0.952 | 0.171 | 0.222 | **2.526** | **0.953** |
| | VMD+LSTM | 0.096 | 0.127 | 4.795 | **0.957** | **0.166** | **0.218** | 2.527 | 0.951 |
| | LSTM | 0.224 | 0.302 | 7.417 | 0.892 | 0.326 | 0.479 | 4.269 | 0.848 |

In the SWH forecasting at four buoys, the buoy with the best performances was buoy 51003 with MAE, RMSE, MAPE and $R^2$ of 0.066 m, 0.082 m, 2.315 %, and 0.978, respectively. Among the APD forecasting at four buoys, the VMD-TCN-LSTM



model had the most petite MAE and RMSE at buoy 51003, with 0.105 s and 0.145 s, respectively, and the smallest MAPE and the highest $R^2$ at buoy 51004 with 1.323 % and 0.981, respectively.

To compare the forecasting results of different models more visually, Figure 7 shows the comparison results of the 3-hour
SWH and APD forecasting curves of different models with the observed values for the first 24 hours of the testing set for each buoy. As shown in Fig. 7, the forecasting results of VMD-TCN-LSTM have a good agreement with the observed values of NDBC at most moments on all four buoys. The forecasting results of VMD-LSTM are also close to the observed values. Meanwhile, the results of both the VMD-TCN-LSTM and VMD-LSTM models are significantly better than those of the LSTM model. It shows that both VMD-TCN-LSTM and VMD-LSTM models can better capture the time-varying
characteristics of wave series data and thus perform well in the SWH and APD forecasts.



**Figure 7. Comparison results of the 3-hour SWH and APD forecasting curves of different models with the observed values for the first 24 hours of the testing datasets for each buoy.**

Figure 8 shows the linear fitting results of the SWH and APD observations with the forecasts of the three models for each
buoy. According to the linear fitting formula, the fitting curves of both the VMD-LSTM and VMD-TCN-LSTM models
were closer to "$y = x$" compared to the LSTM model. For the 3-hour SWH forecasts, the fitted formula of the VMD-TCN-
LSTM forecasting results for buoy 51004 was closest to "$y = x$", which had a slope of 0.9817 and an intercept of 0.0404 (Fig.
8(e)). For the 3-hour APD forecasts, the fitted formula of the VMD-TCN-LSTM forecasting results for buoy 51004 was
closest to "$y = x$", which had a slope of 0.9929 and an intercept of 0.0829 (Fig. 8(f)). The results indicate that the forecasting
performance of these two models is significantly better than that of the LSTM model, which is consistent with the findings in
Fig. 7 and Table 5.

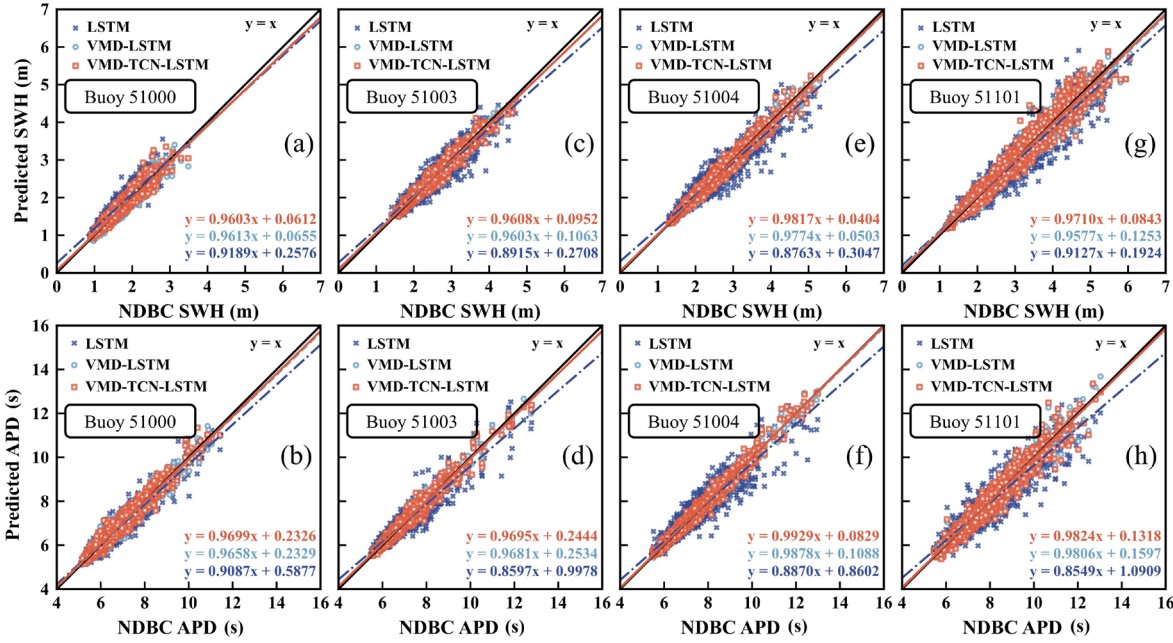

**Figure 8. The linear fitting of the 3-hour SWH and APD predictions and observations for the three models.**

Meanwhile, the SWH and APD of the four buoys have different ranges of values and other statistical features, which proves
that the two models, VMD-LSTM and VMD-TCN-LSTM, have good robustness for SWH and APD forecasting under
different scenarios. The VMD technique can extract the time-varying features of the original data, contributing to the
accuracy of the prediction model. In addition, using TCN cells instead of LSTM cells for encoding the network model can
also reduce the error of the prediction model by a small amount.

**5.3 12-hour forecasting performance**

The evaluation metrics of SWH and APD for different prediction models on the testing sets of the four buoys for the 12-hour
forecasts are shown in Table 6, and the best results are shown in bold in the table. As shown in Table 6, both the VMD-



LSTM and VMD-TCN-LSTM models significantly outperform the performances of the LSTM model. This is like the results of the 3-hour SWH and APD forecasts.

In addition, the performances of the VMD-TCN-LSTM model outperformed the VMD-LSTM for the SWH and APD forecasts at all buoys. Compared with the 3-hour forecasts, the TCN cells were more significant for the model performance improvement in the 12-hour wave forecasts. This is because the residual block structure used in the TCN cells can maintain the historical information for a long time. The TCN cells are more significant in the longer time wave parameter forecasts.

Among the SWH forecasting of the four buoys, the VMD-TCN-LSTM model had the smallest MAE and RMSE at buoy 51000 with 0.125 m and 0.165 m, respectively. Buoy 51003 had the smallest MAPE of 5.912 %. Buoy 51004 had the largest $R^2$ of 0.898. In the APD forecasting at four buoys, the VMD-TCN-LSTM model had the most petite MAE and RMSE at buoy 51003, with 0.247 s and 0.336 s, respectively, and the smallest MAPE and the highest $R^2$ at buoy 51004 with 3.329 % and 0.904, respectively.

The comparison of the forecasting curves of different models with the observations of NDBC for the first 24 hours of the testing set of the four NDBC buoys for the 12-hour SWH and APD forecasts is shown in Fig. 9. As shown in the figure, the forecasts of the VMD-TCN-LSTM model were in excellent agreement with the NDBC observations for most moments at all four buoys. And it is significantly outperforming the forecasting curves of VMD-LSTM and LSTM models. The results show that the VMD-TCN-LSTM model can better capture the time-varying characteristics of wave series data and thus performs well in forecasting SWH and APD.

**Table 6. Accuracy evaluation of the three models in 12-hour SWH and APD forecasts.**

| Buoy ID | Algorithm | SWH | | | | APD | | | |
|---|---|---|---|---|---|---|---|---|---|
| | | MAE (m) | RMSE (m) | MAPE (%) | $R^2$ | MAE (s) | RMSE (s) | MAPE (%) | $R^2$ |
| 51000 | VMD+TCN+LSTM | **0.125** | **0.165** | **7.128** | **0.817** | **0.282** | **0.382** | **4.212** | **0.834** |
| | VMD+LSTM | 0.136 | 0.182 | 7.706 | 0.772 | 0.302 | 0.406 | 4.520 | 0.801 |
| | LSTM | 0.195 | 0.248 | 11.62 | 0.639 | 0.353 | 0.485 | 5.210 | 0.710 |
| 51003 | VMD+TCN+LSTM | **0.152** | **0.203** | **5.912** | **0.872** | **0.247** | **0.336** | **3.355** | **0.871** |
| | VMD+LSTM | 0.177 | 0.233 | 6.910 | 0.830 | 0.293 | 0.398 | 3.948 | 0.819 |
| | LSTM | 0.271 | 0.371 | 10.39 | 0.581 | 0.439 | 0.629 | 5.862 | 0.550 |
| 51004 | VMD+TCN+LSTM | **0.157** | **0.206** | **6.167** | **0.898** | **0.259** | **0.361** | **3.329** | **0.904** |
| | VMD+LSTM | 0.169 | 0.222 | 6.575 | 0.882 | 0.291 | 0.395 | 3.799 | 0.884 |
| | LSTM | 0.277 | 0.398 | 10.65 | 0.619 | 0.506 | 0.743 | 6.457 | 0.581 |
| 51101 | VMD+TCN+LSTM | **0.283** | **0.369** | **9.575** | **0.839** | **0.419** | **0.555** | **5.262** | **0.811** |
| | VMD+LSTM | 0.274 | 0.361 | 9.270 | 0.840 | 0.441 | 0.585 | 5.526 | 0.787 |
| | LSTM | 0.384 | 0.522 | 12.98 | 0.673 | 0.638 | 0.918 | 7.876 | 0.450 |



**Figure 9. Comparison results of the 12-hour SWH and APD forecasting curves of different models with the observed values for the first 24 hours of the testing datasets for each buoy.**



Figure 10 shows the linear fitting results for the 12-hour SWH and APD forecasts data and observations at each buoy for the
three models. As shown in Fig. 10, it was evident that the forecasting results of the VMD-TCN-LSTM model have the closest fitting formula to "$y = x$" compared with the LSTM model, and the VMD-TCN-LSTM model is better than the VMD-LSTM model. In the 12-hour SWH forecasts, the fitted formula of the VMD-TCN-LSTM forecasting results for buoy 51000 was closest to "$y = x$", which had a slope of 0.9256 and an intercept of 0.1252 (Fig. 10(a)). Among the 12-hour APD forecasts, the fitted formula of the VMD-TCN-LSTM forecasting results for buoy 51004 was closest to "$y = x$", which had a
slope of 0.9664 and an intercept of 0.2500 (Fig. 10(f)). Both VMD-TCN-LSTM and VMD-LSTM models have significantly better forecasting performance than the LSTM model. This is consistent with the conclusions of Fig. 9 and Table 6.

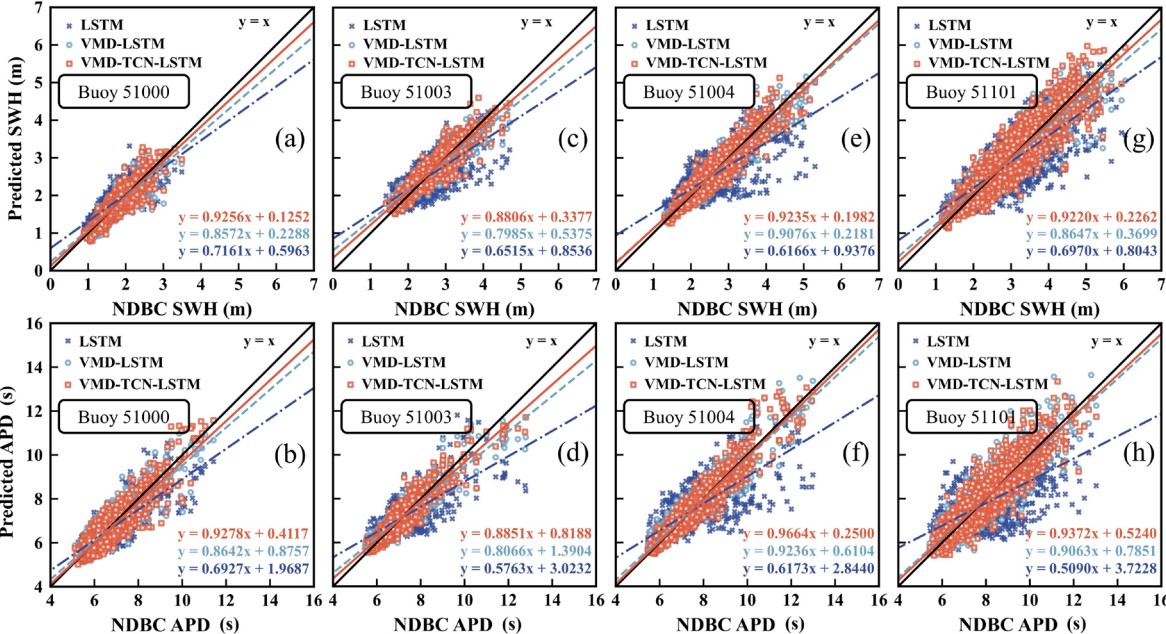

**Figure 10. The linear fitting of the 12-hour SWH and APD predictions and observations for the three models.**

Moreover, the variability of the numerical ranges of SWH and APD for the four buoys also demonstrates the excellent
robustness of the VMD-TCN-LSTM model for SWH and APD forecasts in different scenarios. The pre-processing of wave sequence data using VMD can extract the time-varying features of the original data well, and the expansion convolution module of TCN increases the perceptual field of the model. At the same time, the residual block enables the preservation of the long-term information of the original data. Therefore, the hybrid model of VMD, TCN, and LSTM can significantly improve the accuracy of the forecasting results.



## 5.4 24-, and 48-hour forecasting performance


To further compare the performance of the VMD-TCN-LSTM model for the longer time wave forecasts, the error indices of the prediction models at 24 and 48 hours are presented in Table 7 and Table 8, respectively, where the best results are shown in bold in the table.

**Table 7. Accuracy evaluation of the three models in 24-hour SWH and APD forecasts.**

| Buoy ID | Algorithm | SWH | | | | APD | | | |
|---|---|---|---|---|---|---|---|---|---|
| | | MAE (m) | RMSE (m) | MAPE (%) | $R^2$ | MAE (s) | RMSE (s) | MAPE (%) | $R^2$ |
| 51000 | VMD+TCN+LSTM | **0.119** | **0.173** | **8.203** | **0.733** | **0.302** | **0.412** | **4.287** | **0.775** |
| | VMD+LSTM | 0.149 | 0.206 | 8.406 | 0.693 | 0.336 | 0.459 | 4.976 | 0.739 |
| | LSTM | 0.249 | 0.313 | 14.87 | 0.294 | 0.464 | 0.648 | 6.841 | 0.478 |
| 51003 | VMD+TCN+LSTM | **0.194** | **0.247** | **7.604** | **0.808** | **0.312** | **0.420** | **4.268** | **0.810** |
| | VMD+LSTM | 0.233 | 0.290 | 9.152 | 0.734 | 0.342 | 0.457 | 4.705 | 0.764 |
| | LSTM | 0.381 | 0.503 | 14.44 | 0.302 | 0.585 | 0.842 | 7.777 | 0.298 |
| 51004 | VMD+TCN+LSTM | **0.191** | **0.253** | **7.408** | **0.845** | **0.337** | **0.467** | **4.266** | **0.833** |
| | VMD+LSTM | 0.213 | 0.282 | 8.302 | 0.808 | 0.405 | 0.555 | 5.297 | 0.764 |
| | LSTM | 0.362 | 0.519 | 14.29 | 0.349 | 0.693 | 0.959 | 8.941 | 0.295 |
| 51101 | VMD+TCN+LSTM | **0.309** | **0.400** | **10.75** | **0.803** | **0.496** | **0.671** | **6.258** | **0.688** |
| | VMD+LSTM | 0.325 | 0.416 | 11.49 | 0.787 | 0.517 | 0.701 | 6.497 | 0.659 |
| | LSTM | 0.578 | 0.780 | 18.81 | 0.247 | 0.847 | 1.169 | 10.43 | 0.257 |


**Table 8. Accuracy evaluation of the three models in 48-hour SWH and APD forecasts.**

| Buoy ID | Algorithm | SWH | | | | APD | | | |
|---|---|---|---|---|---|---|---|---|---|
| | | MAE (m) | RMSE (m) | MAPE (%) | $R^2$ | MAE (s) | RMSE (s) | MAPE (%) | $R^2$ |
| 51000 | VMD+TCN+LSTM | **0.187** | **0.249** | **10.64** | **0.551** | **0.443** | **0.604** | **6.676** | **0.487** |
| | VMD+LSTM | 0.197 | 0.261 | 10.87 | 0.505 | 0.476 | 0.656 | 6.899 | 0.432 |
| | LSTM | 0.312 | 0.390 | 19.06 | 0.204 | 0.602 | 0.798 | 8.851 | 0.160 |
| 51003 | VMD+TCN+LSTM | **0.315** | **0.387** | **12.56** | **0.536** | **0.448** | **0.606** | **6.174** | **0.592** |
| | VMD+LSTM | 0.335 | 0.428 | 13.51 | 0.434 | 0.531 | 0.792 | 7.110 | 0.429 |
| | LSTM | 0.552 | 0.720 | 19.85 | 0.214 | 0.772 | 1.097 | 10.41 | 0.214 |
| 51004 | VMD+TCN+LSTM | **0.255** | **0.339** | **9.879** | **0.723** | **0.524** | **0.715** | **6.714** | **0.611** |
| | VMD+LSTM | 0.299 | 0.389 | 11.78 | 0.635 | 0.564 | 0.787 | 7.247 | 0.529 |
| | LSTM | 0.469 | 0.644 | 18.71 | 0.231 | 0.859 | 1.243 | 11.08 | 0.276 |
| 51101 | VMD+TCN+LSTM | **0.456** | **0.586** | **16.23** | **0.580** | **0.744** | **0.907** | **9.474** | **0.432** |
| | VMD+LSTM | 0.497 | 0.648 | 16.74 | 0.487 | 0.822 | 1.109 | 10.31 | 0.390 |
| | LSTM | 0.651 | 0.805 | 23.86 | 0.238 | 1.127 | 1.503 | 13.65 | 0.180 |



As shown in Table 7,, for the 24-hour forecasts, the MAE and RMSE for the forecasting of SWH and APD at buoy 51000 are the minimum, with MAE of 0.119 m and 0.302 s, and RMSE of 0.173 m and 0.412 s, respectively. This is because the range of data for SWH and APD in the testing datasets at buoy 51000 is the minimum (Fig. 2). Buoy 51004, the forecasting of SWH and APD had the minimum MAPE and the maximum $R^2$, with MAPE of 7.408 % and 4.266 %, and $R^2$ of 0.845 and 0.833, respectively.

As shown in Table 8, for the 48-hour forecasts, the MAE and RMSE for the forecasting of SWH and APD at buoy 51000 are the minimum, with MAE of 0.187 m and 0.443 s and RMSE of 0.249 m and 0.604 s, respectively. It showed a similar performance as the 24-hour SWH and APD forecasts. Buoy 51004 had the maximum $R^2$ with 0.723 and 0.611 for SWH and APD forecasts, respectively. Buoy 51004 also had a minimum MAPE of 9.879 % for the SWH forecasts. Buoy 51003 had a minimum MAPE of 6.174 % for the APD forecasts.

**5.5 Analysis of improvement of VMD-TCN-LSTM compared with previous models**

To precisely quantify the prediction performance improvement rate of the VMD technique and TCN cells for the LSTM model, respectively. The model performance improvement rates for VMD-TCN-LSTM and VMD-LSTM were calculated by using Eqs. (24) to (27) (Table 9), and bold in the table represents the highest result of the model performance improvement rate. As shown in Table 9, VMD-LSTM and VMD-TCN-LSTM models had very similar improvement rates in MAE, RMSE, MAPE, and $R^2$ in the 3-hour SWH forecasts, which indicates that the improvement of the VMD-TCN-LSTM model for prediction accuracy in the 3-hour SWH forecasts is mainly contributed by the VMD technique. The same conclusion can be obtained in the 3-hour APD forecasts. Subsequently, when the length of forecasting increases to 12, 24, and 48 hours, the TCN cells is more significant for the decrease of MAE, RMSE, MAPE, and the increase of $R^2$ for the forecasting results.

**Table 9. The performance improvement rate of VMD-TCN-LSTM and VMD-LSTM models relative to LSTM model.**

| Evaluation indicators | Algorithm | SWH | | | | APD | | | |
|---|---|---|---|---|---|---|---|---|---|
| | | 3-hour | 12-hour | 24-hour | 48-hour | 3-hour | 12-hour | 24-hour | 48-hour |
| $I_{MAE}$ (%) | VMD+TCN+LSTM | **51.75** | **37.36** | **48.77** | **39.65** | **48.47** | **36.75** | **43.60** | **35.34** |
| | VMD+LSTM | 50.74 | 33.14 | 40.98 | 34.02 | 46.27 | 30.27 | 37.41 | 28.39 |
| $I_{RMSE}$ (%) | VMD+TCN+LSTM | **51.91** | **39.08** | **48.90** | **39.24** | **50.51** | **39.69** | **45.11** | **37.80** |
| | VMD+LSTM | 49.71 | 34.72 | 42.22 | 33.18 | 49.07 | 34.03 | 39.26 | 27.12 |
| $I_{MAPE}$ (%) | VMD+TCN+LSTM | **48.98** | **37.52** | **45.80** | **40.02** | **47.59** | **35.89** | **43.69** | **33.82** |
| | VMD+LSTM | 47.12 | 33.51 | 40.23 | 35.45 | 45.55 | 29.22 | 36.31 | 28.20 |
| $I_{R^2}$ (%) | VMD+TCN+LSTM | **8.733** | **36.92** | **171.0** | **169.3** | **10.63** | **52.91** | **146.0** | **160.6** |
| | VMD+LSTM | 8.560 | 32.74 | 157.2 | 132.5 | 10.30 | 47.19 | 131.6 | 119.7 |

There was no significant rule for the decreased rate of TCN cells on the MAE, RMSE, and MAPE of the model at various forecasting time length. However, the contribution of TCN cells to the improvement of $R^2$ for forecasting results gradually



increases with the increase of forecasting time length. It reaches the maximum value in the 48-hour SWH and APD forecasts. As shown in Table 9, in the 48-hour SWH forecasts, the VMD technique increases the $R^2$ of the forecasting performance by 132.5 %, and the TCN cells for model encoding resulted in a further 36.8 % improvement in the $R^2$ of the model. In the 48-hour APD forecasts, the VMD technique increases the $R^2$ of the forecasting performance by 119.7 %. The TCN cells resulted in a further 40.9 % improvement in the $R^2$ of the model.

## 6 Conclusions

This paper proposes a hybrid VMD-TCN-LSTM model for forecasting SWH and APD using buoy data near the Hawaiian Islands provided by the NDBC. Seven physical parameters, SWH, APD, MWD, WSPD, WDIR, PRES, and ATMP, were chosen for training the prediction model in the research. Specifically, the original features of the non-smooth wave series data were extracted by decomposing the original SWH and APD series data using the VMD technique. Subsequently, a

prediction model is constructed using a network structure encoded by TCN cells and decoded by LSTM cells, where the TCN cells can capture the local feature information of the original series and can maintain the historical information for a long time. Simultaneously, the BO algorithm is used to obtain the optimal hyperparameters of the model to prevent overfitting or underfitting problems of the model. Ultimately, the 3-, 12-, 24-, and 48-hour forecasts of SWH and APD were implemented based on the VMD-TCN-LSTM model. In addition, eight evaluation metrics, MAE, RMSE, MAPE, $R^2$, $I_{MAE}$,

$I_{RMSE}$, $I_{MAPE}$, and $I_{R^2}$, were used to evaluate and test the model performance.

The VMD-TCN-LSTM model proposed in this research outperforms the LSTM and the VMD-LSTM models for all forecasting time lengths at all four NDBC buoys. It demonstrates that the VMD-TCN-LSTM model has good robustness and generalization ability. In the 3-hour SWH and APD forecasts, the improvement of the hybrid model for forecasting accuracy is mainly contributed by the VMD technique, and the contribution of the TCN cells to the advancement of the model

accuracy is relatively tiny. Subsequently, the contribution of TCN cells to improve model forecasting accuracy was gradually significant when the forecasting time length increased to 12, 24, and 48 hours.

There was no significant rule for the decreased rate of TCN cells on the MAE, RMSE, and MAPE of the model at various forecasting time lengths. The contribution of TCN cells to improving $R^2$ for forecasting results gradually increases with the increase of forecasting time length. The VMD technique and the TCN cells improved the $R^2$ of the model by 132.5 % and

36.8 %, respectively, in the 48-hour SWH forecasts. In the 48-hour APD forecasts, the VMD technique and the TCN cells improved the $R^2$ of the model by 119.7 % and 40.9 %, respectively.

**Data availability**

The buoy data can be found in the National Data Buoy Center (https://www.ndbc.noaa.gov/, last access: 5 June 2022).



**Author contributions**

QJ and LH provided the initial scientific idea, conducted the models' experiments, and wrote the original paper. LJ and YL supervised the work. YZ and MX collected and pre-processed the data sets. All authors reviewed and edited the manuscript to its final version.

**Competing interests**

The authors declare that they have no conflict of interest.

**Acknowledgements**

We thank the wave buoy data obtained from the National Data Buoy Center for the case study in this paper. We also thank the Key Laboratory of South China Sea Meteorological Disaster Prevention and Mitigation of Hainan Province, the Basic Scientific Research Business Expenses of Zhejiang Provincial Universities, the Basic Public Welfare Research Project of Zhejiang Province, and the Innovation Group Project of Southern Marine Science and Engineering Guangdong Laboratory

(Zhuhai) for funding and supporting this research.

**Financial support**

This research was funded by the Key Laboratory of South China Sea Meteorological Disaster Prevention and Mitigation of Hainan Province (grant no. SCSF202204), the Basic Scientific Research Business Expenses of Zhejiang Provincial Universities (grant no. 2020J00008), the Basic Public Welfare Research Project of Zhejiang Province (grant no.

LGF22D060001), and the Innovation Group Project of Southern Marine Science and Engineering Guangdong Laboratory (Zhuhai) (grant no. SML2020SP007 and 311020004).

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
