# Peer review of "Short-term Prediction of the Significant Wave Height and Average Wave Period based on VMD-TCN-LSTM Algorithm"

_EGUsphere, 2023_

## Author Comment (AC1)

**Response to Reviewer 1**

Many thanks for reviewing our manuscript. Your comments and suggestions are very helpful and constructive. We have addressed all of your concerns in the revised manuscript.

The issues proposed by the reviewers have all been addressed.

**Please notice:**

Reviewer comments in quotations; our responses in blue;

Main changes are:

Comments and Suggestions for Authors

Considering the need to enhance predictions of ocean wave parameters, Ji et al. considered the adoption of a joint VMD-TCN-LSTM algorithm to forecast significant wave height and wave period with minor computational expense and using direct buoy observations. The paper is of use to the community and while I have no technical objections, there are, however, a few issues the authors should consider. They are as follows:

1. The introduction of the model and its settings is very detailed but far too long as it consumes the first 13 pages and 6 pages of the article. These can be either reduced significantly or placed within a supplement to join the manuscript. This will allow for readers to focus on the results section which should be the manuscript's centerpiece.

Reply:

1) We gratefully appreciate for your valuable comment.

2) We have simplified the model and its settings as suggested. We have given the detail information in the supplement.

On L128-L143, we have simplified the temporal convolutional networks model. On L157-L171, we have simplified the Bayesian optimization algorithm. We have given the detail information of the variational mode decomposition (VMD) algorithm and long short-term memory networks model in Appendix A and B, respectively. On L185-L187, we have simplified the parameter settings of the VMD algorithm.

2. On L32, the reference (P. et al., 2020) does not follow the format of the other references. Please revise.

Reply:

1)   Thank you so much for your careful check. We are very sorry for our carelessness.

2)   On L32, the reference (P. et al., 2020) revised to (Pushpam P. and Enigo V.S., 2020).

3. There is a space missing on L47 before Zhao et al., 2019.

Reply:

1)   Thank you so much for your careful check. We are very sorry for our carelessness.

2)   On L47, we added a space before the Zhao et al., 2019.

4. I don't understand why the SST or water temperature would directly affect wave activity. Indeed

WTMP and ATMP are negatively correlated with wave parameters in Figure 3. Please justify the usage of these variables and check if the forecast skill improves with their addition/subtraction in a new experiment. If forecast skill does not change with their removal, you'll have your answer on if its necessary to include it them in an already extensive list of predictands.

Reply:

1) We gratefully appreciate for your valuable comment.

2) We agree that the WTMP and ATMP do not directly affect the wave, physically. The ocean wave height and average wave period are mainly influenced by wind direction, wind speed, air pressure, previous wave height and wave period.

The wave prediction method used in this study is a data-driven method. Therefore, some variables that are not physically causally significant may also influence the prediction results. We focus on how VMD and TCN affect LSTM's ability to predict wave, without considering the effect of the data itself on the prediction skill. Using similar data-driven approaches (GRU network, LSTM) for wave prediction, some studies include temperature as one of the driving data (Li et al., 2022), but others do not (Fan et al., 2020). The suggestion is a good way to test the effect of SST or even all variables on the prediction skill. We did not carry out relevant experiments, but it will be carefully considered in future studies.

Li, X., Cao, J., Guo, J., Liu, C., Wang, W., Jia, Z., and Su, T.: Multi-step forecasting of ocean wave height using gate recurrent unit networks with multivariate time series, Ocean Eng., 248, 110689, https://doi.org/10.1016/j.oceaneng.2022.110689, 2022.

Fan, S., Xiao, N., and Dong, S.: A novel model to predict significant wave height based on long short-term memory network, Ocean Eng., 205, 107298, https://doi.org/10.1016/j.oceaneng.2020.107298, 2020.

5. The range of APD in Table 2 and in Figure 2 seem to indicate the occurrence of both wind waves and swell. Were wave forecasts done on both systems together, or individually? As swell is generally insensitive to wind information, using wind to predict swell may be ineffective.

Reply:

1) We gratefully appreciate for your valuable comment.

2) Table 2 and Figure 2 show that wind waves and swells occur simultaneously. Since the buoy data contains both wind and swell information, we use the buoy data directly as training data, and thus the predicted waves contain both wind waves and swells. Although swell is insensitive to wind, it is difficult to distinguish wind waves from swell from observed buoy data. With this data, our forecast models cannot predict wind waves and swell separately as the wave models such as SWAN or WAVEWATCH III do.

6. There should be a colon (:) instead of a period (.) at the end of the sentence on L130. Same for L206.

Reply:

1) We gratefully appreciate for your valuable comment.

2) On L130 and L206, the period (.) at the end of the sentence revised to colon (:).

7. There is a duplication of a comma on L372 after Table 7.

Reply:
1) Thank you so much for your careful check. We are very sorry for our carelessness.
2) On L372, we removed the duplicate comma.

8. It might be useful in the conclusion to discuss the implications of the research on, for example, ocean wave energy projects that would be heavily dependent on wave height and period forecasts.
Reply:
1) We gratefully appreciate for your valuable comment.
2) Implications and future work are discussed in the conclusion, which can be found in the last paragraph of the discussion.

Now that the short term SWH and APD can be accurately predicted using the hybrid VMD-TCN-LSTM, this method would be useful for some marine related activities which are highly dependent on wave height and period predictions, such as ocean wave energy projects, shipping, fishing, coastal structures, and naval operations. Future work will investigate the effect of different driving data on the prediction skill, or the use of VMD-TCN-LSTM to predict other marine environmental parameters (e.g., sea level or winds). The combination of numerical wave models and the VMD-TCN-LSTM for large-scale SWH and APD simulations will also be developed.

The manuscript has been revised carefully according to the reviewer's comments. We are appreciated for the reviewer's constructive comments and kind help. The quality of the revised manuscript has been improved under the help of the reviewer, and hope that the correction will meet with approval.
Once again, thank you very much for your comments and suggestions.

Yours sincerely.

Manuscript title: Short-term Prediction of the Significant Wave Height and Average Wave Period based on VMD-TCN-LSTM Algorithm. (egusphere-2023-960)

Authors: Qiyan Ji, Lei Han, Lifang Jiang, Yuting Zhang, Minghong Xie, and Yu Liu

Correspondence:

Lei Han

s20070700026@zjou.edu.cn

---

## Author Comment (AC2)

**Response to Anonymous Referee #2**

Many thanks for reviewing our manuscript. Your comments and suggestions are very helpful and constructive. We have addressed all of your concerns in the revised manuscript.

The issues proposed by the reviewers have all been addressed.

**Please notice:**

Reviewer comments in quotations; our responses in blue;

Main changes are:

Comments and Suggestions for Authors

Wave prediction is very important for fisheries, wave power generation and marine transportation. Numerical modelling (e.g., SWAN or WAVEWATCHIII model) is a common method for operational wave forecasting. Data-driven methods, such as neural network methods, are also very popular. This paper proposed a hybrid VMD-TCN-LSTM model to forecast significant wave height and wave period. The results show that the method is effective in predicting ocean waves. However, some issues need to be clarified.

1. The descriptions of the VMD, TCN and LSTM methods are very detailed. As these methods are widely used in other fields, the corresponding description can focus more on the improvement of these methods in this study.

Reply:

1) We gratefully appreciate for your valuable comment.

2) In the revised manuscript, we have simplified the description of the methodology, and the detail information the methods are migrated to the appendix.

3) On Lines 128-143, we have simplified the temporal convolutional networks model. On Lines 157-171, we have simplified the Bayesian optimization algorithm. We have given the detail information of the variational mode decomposition (VMD) algorithm and long short-term memory networks model in Appendix A and B, respectively. On Lines 185-187, we have simplified the parameter settings of the VMD algorithm.

2.In situ measurements from four buoys were used in this study. Does the hybrid VMD-TCN-LSTM wave prediction model use the same parameters measured at these buoy stations?

Reply:

1) Thank you for the question.

2) In this study, we use the Bayesian optimization algorithm for the hyperparameter finding of the VMD-TCN-LSTM wave prediction model. In the four buoys, we used the same parameters.

3.Line 104~105: The GST has a positive relation with SWH. Why not use this physical parameter to drive the model?

Reply:

1) We gratefully appreciate for your comment.

2) As you mentioned, The GST has a positive relation with SWH. However, as shown in Figure 3, the correlation between the GST (gust speed) and the WSPD (wind speed) is as high as 0.988, the WSPD and the GST have very similar characteristics of variation. Therefore, in order to reduce the redundancy of the input data to the forecast model, we just choose WSPD to represent the temporal variation of wind.

4. Line 211~212: The BO has two critical components. First, establish an agency model of the objective function through a regression model (e.g., Gaussian process regression) and subsequently use the acquisition function to decide where to sample next (Frazier, 2018). The word "build" and "use" should be revised as "establishing" and "using".

Reply:

1) We gratefully appreciate for your valuable comment.

2) In Lines 161~163 of the revised manuscript. we corrected the words as you suggested.

5. Line 628: "To quantify the prediction model performance" should be revised as "To quantify the performance of the prediction model".

Reply:

1) Thank you for your comment.

2) The first submitted version of the manuscript did not have line 628, perhaps you are referring to line 267 of the manuscript, which we changed in the revised version of the manuscript at line 210, as you suggested.

6. Line 294: "in 3-hour SWH and APD forecasts" and Line 414 "In the 3-hour SWH and APD forecasts". The word "in" should be revised as "for".

Reply:

1) Thank you so much for your careful check.

2) In Lines 232, 234 and 371 of the revised manuscript. we corrected the words as you suggested.

7. Line 375: Please add "at" before "Buoy 51004".

Reply:

1) Thank you so much for your careful check.

2) In Line 322 of the revised manuscript. we add "at" before "Buoy 51004" as you suggested.

8. Line 391: "the TCN cells is." Here, "is" should be "as".

Reply:

1) Thank you so much for your careful check.

2) In Line 337 of the revised manuscript. we corrected the "is" to "as" as you suggested.

The manuscript has been revised carefully according to the reviewer's comments. We are appreciated for the reviewer's constructive comments and kind help. The quality of the revised manuscript has been improved under the help of the reviewer, and hope that the correction will meet with approval.

Once again, thank you very much for your comments and suggestions.

Yours sincerely.

Manuscript title: Short-term Prediction of the Significant Wave Height and Average Wave Period based on VMD-TCN-LSTM Algorithm. (egusphere-2023-960)

Authors: Qiyan Ji, Lei Han, Lifang Jiang, Yuting Zhang, Minghong Xie, and Yu Liu

Correspondence:

Lei Han

s20070700026@zjou.edu.cn

---

## Author Comment (AC3)

**Response to Anonymous Referee #3**

Many thanks for reviewing our manuscript. Your comments and suggestions are very helpful and constructive. We have addressed all of your concerns in the revised manuscript.

The issues proposed by the reviewers have all been addressed.

**Please notice:**

Reviewer comments in quotations; our responses in blue;

Main changes are:

Comments and Suggestions for Authors

The manuscript proposes a VMD-TCN-LSTM hybrid model to predict significant wave height and average wave period. The theoretical innovation of this article is not remarkable. However, before considering publishing in this top journal, this study lacks an in-depth comparative analysis of the data. The issues listed below should be addressed by the authors.

1. In the introduction, much more references related with wave period prediction are expected to cite for overall literature review.

Reply:

1) We gratefully appreciate for your valuable comment.

2) Wave prediction models based on machine learning are more likely to predict wave height and less likely to predict period. There have been a few studies in recent years that have attempted to predict both wave height and period. We added some explanation and references related with wave period prediction. In Line 59-61.

For example, Hu et al. (2021) used XGBoost and LSTM to forecast wave heights and periods. Based on multi-layer perceptron and decision tree architecture, Luo et al. (2023) realized the prediction of effective wave height, average wave period, and average wave direction.

Hu, H., van der Westhuysen, A. J., Chu, P., and Fujisaki-Manome, A.: Predicting Lake Erie wave heights and periods using XGBoost and LSTM, Ocean Model., 164, 101832, https://doi.org/10.1016/j.ocemod.2021.101832, 2021.

Luo, Y., Shi, H., Zhang, Z., Zhang, C., Zhou, W., Pan, G., and Wang, W.: Wave field predictions using a multi-layer perceptron and decision tree model based on physical principles: A case study at the Pearl River Estuary, Ocean Eng., 277, 114246, https://doi.org/10.1016/j.oceaneng.2023.114246, 2023.

2. It is recommended to set a threshold to distinguish whether the center frequency has changed significantly.

Reply:

1) We gratefully appreciate for your valuable comment.

2) In Line 190 of the revised manuscript, we set the change threshold of the center frequency set to

1e-8 Hz to distinguish whether the center frequency has changed significantly.

3. Have other wave parameters such as MWD or WSPD been decomposed by VMD for prediction? If they are decomposed, please add their K values, otherwise explain the parameter composition of input.
Reply:
1) We gratefully appreciate for your valuable comment.
2) Other wave parameters are not required to be decomposed by VMD. In Lines 196-198 of the revised manuscript, we add the explanation of the input parameters to the model.
3) The input parameters to the model includes 13 SWH IMFs and residual, 12 APD IMFs and residual, original MWD, WSPD, PRES and ATMP, recoded WDIR.

4. Are the hyper-parameter optimization results in Table 4 obtained from these search intervals? Are they obtained from search spaces containing several specific values? Much more explanation are suggested to provide.
Reply:
1) We gratefully appreciate for your valuable comment.
2) The hyper-parameter optimization results in Table 4 obtained from search set spaces containing several specific values. We have revised Table 4 for a clearer explanation.

5. What's the maximum epochs set for each model during training?
Reply:
1) Thank you for your comment.
2) The maximum epochs set for each model during training is 500. Meanwhile, we use the Early Stopping method to reduce the wasted training time, so the final value of the epochs of each model will be less than 500.

6. Please check all bold metrics values. It seems that the MAE, RMSE, MAPE and R2 of VMD-TCN-LSTM in SWH prediction at 51101 given in Table 6 are not the best.
Reply:
1) Thank you so much for your careful check. We are very sorry for our carelessness.
2) We have corrected the bold metrics values in Table 6 on the revised manuscript.

7. What are the lags of each input variable chosen for prediction?
Reply:
1) Thank you for your question.
2) The lags of each input variable chosen for prediction are 3 hours. in Line 198 of the revised manuscript., we add the explanation.

8. Compared with previous methods, the properties of the proposed method should be summarized to describe clear findings of this study.
Reply:
1) We gratefully appreciate for your valuable comment.
2) We add some discussion in the section 5.5 line 348~357 to summarize the properties of the

proposed method.

LSTM has advantages in solving the prediction problem by using time series data, and has been widely used in many fields. However, due to the strong nonlinear effects in the generation and evolution of wave, the wave prediction model that only uses LSTM will weak in the ability of generalization. As a result, both the model's ability to adapt to new samples and its prediction accuracy will be reduced. The VMD signal decomposition method can effectively extract the features of the original wave data, which can enhance LSTM's ability to capture the long-term dependence of the time series data and further improve the performance of the wave prediction model. This study shows that the VDM can significantly reduce the model's MAE, RMSE and MAPE and improve the model's $R^2$. TCN introduces multiple residual blocks to speed up the forecast model and can retain historical wave change information over long periods. This study also shows that TCN's impact increases as the forecast period lengthens. The proposed hybrid VMD-TCN-LSTM shows its advantage in predicting both the wave height and the wave period. This method could also be used in other fields which have similar nonlinear features as waves.

The manuscript has been revised carefully according to the reviewer's comments. We are appreciated for the reviewer's constructive comments and kind help. The quality of the revised manuscript has been improved under the help of the reviewer, and hope that the correction will meet with approval.

Once again, thank you very much for your comments and suggestions.

Yours sincerely.

Manuscript title: Short-term Prediction of the Significant Wave Height and Average Wave Period based on VMD-TCN-LSTM Algorithm. (egusphere-2023-960)

Authors: Qiyan Ji, Lei Han, Lifang Jiang, Yuting Zhang, Minghong Xie, and Yu Liu

Correspondence:

Lei Han

s20070700026@zjou.edu.cn